# The flexible stalk domain of sTREM2 modulates its interactions with brain-based phospholipids

**David Saeb[1], Emma E Lietzke[1,2], Daisy I Fuchs[1], Emma C Aldrich[1], Kimberley D Bruce[2], Kayla G Sprenger[1]***

[1]Department of Chemical and Biological Engineering, University of Colorado Boulder, Boulder, United States; [2]Department of Endocrinology, Metabolism, and Diabetes, University of Colorado Anschutz Medical Campus, Denver, United States

## eLife Assessment

This **useful** manuscript addresses some key molecular mechanisms on the neuroprotective roles of soluble TREM2 in neurodegenerative diseases. The study will advance our understanding of TREM2 mutations, particularly on the damaging effect of known TREM2 mutations, and also provides **solid** evidence why soluble TREM2 can antagonize Aβ aggregation.

**\*For correspondence:**
kayla.sprenger@colorado.edu

**Competing interest:** The authors declare that no competing interests exist.

**Abstract** The microglial surface protein Triggering Receptor Expressed on Myeloid Cells 2 (TREM2) plays a critical role in mediating brain homeostasis and inflammatory responses in Alzheimer's disease (AD). The soluble form of TREM2 (sTREM2) exhibits neuroprotective effects in AD, though the underlying mechanisms remain elusive. Moreover, differences in ligand binding between TREM2 and sTREM2, which have major implications for their roles in AD pathology, remain unexplained. To address these knowledge gaps, we conducted the most computationally intensive molecular dynamics simulations to date of human (s)TREM2, exploring their interactions with key damage- and lipoprotein-associated phospholipids and the impact of the AD-risk mutation R47H. Our results demonstrate that the flexible stalk domain of sTREM2 serves as the molecular basis for differential ligand binding between sTREM2 and TREM2, facilitated by its role in modulating the dynamics of the Ig-like domain and altering the accessibility of canonical ligand binding sites. We identified a novel ligand binding site on sTREM2, termed the 'Expanded Surface 2,' which emerges due to competitive binding of the stalk with the Ig-like domain. Additionally, we observed that the stalk domain itself functions as a site for ligand binding, with increased binding frequency in the presence of R47H. This suggests that sTREM2's neuroprotective role in AD may, at least in part, arise from the stalk domain's ability to rescue dysfunctional ligand binding caused by AD-risk mutations. Lastly, our findings indicate that R47H-induced dysfunction in TREM2 may result from both diminished ligand binding due to restricted complementarity-determining region 2 loop motions and an impaired ability to differentiate between ligands, proposing a novel mechanism for loss-of-function. In summary, these results provide valuable insights into the role of sTREM2 in AD pathology, laying the groundwork for the design of new therapeutic approaches targeting (s)TREM2 in AD.

## Introduction

Alzheimer's disease (AD) is a fatal neurodegenerative condition marked by progressive memory loss, cognitive decline, and impaired daily functioning (*Alzheimer Association, 2024*). Despite extensive research, the root causes remain unknown, and curative treatments remain elusive. AD is hallmarked

**Figure 1.** Overview of structural domains in soluble form of TREM2 (sTREM2) and full-length Triggering Receptor Expressed on Myeloid Cells 2 (TREM2). (**A**) Pre-molecular dynamics (MD) structure of sTREM2$^{WT}$. (**B**) Full sequence of TREM2, indicating significant structural domains.

by the presence of extracellular amyloid-β (Aβ) plaques, intracellular neurofibrillary tau tangles, lipid droplet accumulation, and neuroinflammation (*Kinney et al., 2018*; *Farmer et al., 2020*). Microglia, the brain's macrophages, play a pivotal yet complex role in AD pathology. While their phagocytic activity toward Aβ is considered neuroprotective, they also release cytokines that increase neuroinflammation, ultimately harming neighboring neurons and glia (*Miao et al., 2023*; *Franklin et al., 2021*). Two newly approved monoclonal antibodies targeting Aβ, while transformative, have shown limited efficacy, only modestly slowing disease progression (*Tobeh and Bruce, 2023*). This underscores the need for a deeper understanding of the intricate interactions between the neuroimmune system and AD-relevant proteins.

At the forefront of this investigation is TREM2, a transmembrane receptor expressed on microglial surfaces. TREM2 propagates a downstream signal through interactions with its co-signaling partner, DNAX-activating protein 12 (DAP12). Recent research highlights TREM2's crucial role in modulating microglial responses and maintaining brain homeostasis (*Deczkowska et al., 2020*). TREM2 contains an extracellular immunoglobulin (Ig)-like domain (19–130 amino acid (aa)), a short extracellular stalk domain (131–174 aa), a helical transmembrane domain (175–195 aa), and an intracellular cytosolic tail domain (196–230 aa) (*Dean et al., 2019*). N-linked glycosylation of TREM2 can occur at residues 20 and 79; however its effects are debated (*Park et al., 2016*; *Shirotani et al., 2022*). The Ig-like domain contains several ligand-binding sites. These include a hydrophobic tip, which is characterized by a highly positive electrostatic potential and contains several aromatic residues and three complementarity-determining regions (CDRs), the latter of which (CDR2) also span part of a positively charged basic patch known as the putative ligand interacting region, or 'Surface 1' (*Figure 1*; *Dean et al., 2019*; *Kober et al., 2016*; *Kober et al., 2021*; *Belsare et al., 2022*). Mutations in TREM2 correlate with altered risks of developing AD, with the R47H mutation, found on Surface 1, standing out as a significant genetic risk factor (*Jonsson et al., 2013*; *Guerreiro et al., 2013*; *Abduljaleel et al., 2023*). In vitro and in silico studies have consistently revealed that mutations, including R47H, destabilize TREM2's CDRs (*Dean et al., 2019*; *Menzies et al., 2019*; *Sudom et al., 2018*; *Dash et al., 2022*), exposing once-buried negatively charged residues (*Dean et al., 2019*; *Sudom et al., 2018*)

and disrupting homeostatic TREM2-ligand binding behavior (*Kober et al., 2016*; *Kober et al., 2021*; *Sudom et al., 2018*; *Wang et al., 2015*) .

TREM2 binds diverse anionic and lipidic ligands, including Aβ species (*Kober et al., 2021*; *Zhao et al., 2018*; *Zhong et al., 2018*), lipoproteins (*Kober et al., 2021*; *Bailey et al., 2015*; *Yeh et al., 2016*; *Atagi et al., 2015*), nucleic acids (*Kawabori et al., 2015*), carbohydrates (*Daws et al., 2003*), and phospholipids (PLs) (*Kober et al., 2016*; *Wang et al., 2015*; *Cannon et al., 2012*)—the focus of our study. PLs play a crucial role in lipid metabolism and maintaining brain homeostasis (*Bruce et al., 2017*). TREM2 clears excess PLs during demyelination and interacts with PLs when they are bound to lipoproteins with a core of neutral lipids surrounded by a monolayer of PLs, free cholesterol, and apolipoproteins (*Loving and Bruce, 2020*; *Li et al., 2022*). Many PLs bind to TREM2, including phosphatidyl-choline (PC), -serine (PS), -inositol (PI), -glycerol (PG), -ethanolamine (PE), phosphatidic acid (PA), sphingomyelin (SM), cholesterol, and sulfatide (*Kober et al., 2016*; *Sudom et al., 2018*; *Wang et al., 2015*; *Li et al., 2022*). PLs primarily bind to TREM2's hydrophobic tip and Surface 1, with varying affinities observed in different contexts (*Kober et al., 2016*; *Sudom et al., 2018*; *Wang et al., 2015*). Direct binding assays show stronger TREM2 binding to anionic moieties (PS, PE, PA) and weaker binding to PC and SM. Conversely, TREM2-expressing reporter cells reveal high TREM2 stimulation from PC and SM, especially with the TREM2$^{R47H}$ variant (*Wang et al., 2015*). Collectively, however, these results emphasize TREM2's broad binding capabilities for PLs. Yet, one review suggested effective TREM2 stimulation by PLs may require co-presentation with other molecules, potentially reflecting the nature of lipoprotein endocytosis (*Kober and Brett, 2017*). Another study observed minimal changes in TREM2-PL interactions despite TREM2 mutations (R47H, R62H, T96K) (*Kober et al., 2016*). Ultimately, there remains a major gap in understanding the mechanism by which PLs differentially interact with TREM2 and how these interactions are altered in the presence of disease-associated TREM2 variants.

The activation of TREM2, mediated by the binding of ligands such as PLs, shapes key microglial functions, including proliferation, phagocytosis, and lipid metabolism. Notably, in pathological conditions, ligand-induced TREM2 activation triggers microglial phenotype switching to Disease-Associated Microglia (DAM), characterized by the activation of inflammatory, phagocytic, and lipid metabolic pathways (*Keren-Shaul et al., 2017*). Additionally, TREM2's extracellular domain can undergo cleavage from ADAM 10/17 sheddase at residue H157, yielding soluble TREM2 (sTREM2). The role and relevance of sTREM2 in disease pathology has been heavily debated (*Filipello et al., 2022*). In the cerebrospinal fluid of individuals with early-stage AD, elevated sTREM2 levels have been detected and linked to slower AD progression (*Edwin et al., 2020*; *Franzmeier et al., 2020*; *Ewers et al., 2019*). Furthermore, there is a strong correlation between sTREM2 levels in cerebrospinal fluid and those of Tau, although correlation with Aβ is inconclusive. These findings have established sTREM2 as a long-time biomarker for AD diagnosis and progression (*Suárez-Calvet et al., 2016*; *Suárez-Calvet et al., 2019*; *Park et al., 2021b*).

Many studies have indicated a neuroprotective role for sTREM2 in disease pathology (*Brown and St George-Hyslop, 2022*). It has been suggested, for instance, that sTREM2 may function as a 'dummy receptor' in AD states, preventing disease-associated ligands from binding TREM2 (*Filipello et al., 2022*). Moreover, in vivo AD mouse models evaluating the therapeutic potential of recombinant sTREM2 have observed the suppression of microglial apoptosis, reduced Aβ plaque load, and improved learning and memory abilities (*Zhong et al., 2017*; *Zhong et al., 2019*). More recent studies have indicated that sTREM2 not only serves as an activator for microglial uptake of Aβ but also directly inhibits Aβ aggregation (*Kober et al., 2021*; *Belsare et al., 2022*; *Vilalta et al., 2021*). Specifically, the binding of Aβ to TREM2 has been shown to increase shedding of sTREM2 (*Vilalta et al., 2021*), which can then bind to Aβ oligomers and fibrils to inhibit their secondary nucleation (*Belsare et al., 2022*; *Vilalta et al., 2021*). Interestingly, the effect of R47H on Aβ aggregation is unclear, highlighting the need to study mechanistic aspects of ligand binding (*Belsare et al., 2022*; *Vilalta et al., 2021*).

Some anionic ligands, including Aβ, predominantly bind to Surface 1 on TREM2. Intriguingly, recent observations revealed that Aβ binds to an alternative binding region, termed 'Surface 2,' on sTREM2, situated opposite Surface 1 (*Figure 1*). Surface 2 features a group of positively charged residues surrounded by acidic residues, creating a variegated electrostatic potential (*Belsare et al., 2022*). Herein, we aimed to unravel the molecular basis behind this functionally significant distinction in ligand binding between soluble and membrane-bound TREM2, utilizing molecular dynamics

(MD) simulations. We focused on (s)TREM2-PL interactions, establishing a controlled framework to assess the impacts of various PL chemistries on ligand binding, specifically comparing the binding behavior of anionic PS and neutral PC. We hypothesized that the oft-overlooked flexible stalk domain of sTREM2, minimally explored in previous in silico studies, may play a pivotal role in mediating the observed variations in binding. Furthermore, we sought to understand the impact of the AD-risk mutation R47H on ligand binding, thereby unraveling fundamental roles of (s)TREM2 in AD pathology. To our knowledge, this study represents the second-ever application of MD to investigate sTREM2, totaling an unprecedented 31.2 $\mu s$ of simulation time. Ultimately, this research may unveil new insights into the mechanistic and therapeutic roles of sTREM2 in AD.

## Methods

### Preparation of simulated structures

We employed the AlphaFold (*Varadi et al., 2022*; *Jumper et al., 2021*) model (AF-Q9NZC2-F1) as the initial structure for wild-type (WT) sTREM2 in our simulations, chosen for its inclusion of the unstructured flexible stalk domain. The partial stalk domain spans residues 130 through 157, while the Ig-like domain consists of residues 19 through 130. For WT simulations, we constructed two protein systems: one with just the Ig-like domain ('Ig$^{WT}$') and another containing both the partial stalk and Ig-like domain ('sTREM2$^{WT}$'). Similarly, two protein systems were constructed for the variant simulations: one with just the Ig-like domain containing the R47H mutation ('Ig$^{R47H}$') and another containing both the partial stalk and mutant Ig-like domain ('sTREM2$^{R47H}$'). In contrast to the use of the AlphaFold model for WT, we utilized a crystal structure of TREM2$^{R47H}$ (Protein Data Bank (PDB) code 5UD8 *Sudom et al., 2018*), to which the unstructured stalk domain from the AlphaFold model was added using alignment tools in Visual Molecular Dynamics (VMD). Missing residues were incorporated into the TREM2$^{R47H}$ Ig domain using MODELLER (*Sali and Blundell, 1993*). The initial molecular structures for the PLs, stearoyl-oleoyl-PC (SOPC) and stearoyl-oleoyl-PS (SOPS), were obtained from CHARMM-GUI (*Wu et al., 2014*; *Jo et al., 2009*; *Jo et al., 2007*; *Lee et al., 2019*; *Park et al., 2021a*), with each considered as a singular PL. All eight protein-ligand systems (Ig$^{WT}$, sTREM2$^{WT}$, Ig$^{R47H}$, and sTREM2$^{R47H}$, each with SOPC or SOPS) and six pure-component systems were solvated with explicit water and with counterions added as needed to neutralize the charge.

### Molecular dynamics simulations

All proteins, PLs, and counterions were parameterized using the CHARMM36 force field (*Brooks et al., 2009*; *Lee et al., 2016*), while the TIP3P model was used to describe water (*Bussi et al., 2007*). Prior to subsequent docking studies, each pure-component system underwent steepest-descent energy minimization, initially in vacuum and then in a solvated state. This was followed by a multi-step equilibration protocol, which included a 1 ns NVT equilibration simulation at 310 K using the Bussi-Donadio-Parrinello thermostat (*Bussi et al., 2007*), followed by a 1 ns NPT equilibration simulation at 310 K and 1 bar using the same thermostat and Berendsen barostat (*Berendsen et al., 1984*). Finally, production simulations were carried out at 310 K and 1 bar, utilizing the same thermostat and Parrinello-Rahman barostat (*Parrinello and Rahman, 1981*). The duration of the production simulations was 150 ns for each pure-component PL system and 1 $\mu s$ for each pure-component protein system. Each pure protein system was run with six replicates (see Results).

All simulations were conducted using the GROMACS MD engine (*Abraham et al., 2015*). The LINCS algorithm (*Hess et al., 1997*) was used to constrain bonds involving hydrogen atoms, and particle-mesh Ewald (PME) summations (*Darden et al., 1993*) were employed for calculating long-range electrostatics with a cutoff of 1.2 nm. Lennard-Jones interactions were evaluated up to 1.2 nm and shifted to eliminate energy discontinuities. Neighbor lists were reconstructed in 10-step intervals with a 1.4 nm cutoff. A timestep of 2 fs was implemented in all simulations, and periodic boundary conditions were applied in the x, y, and z directions. Configurations from these production simulations were used as inputs in ensuing docking calculations (see next section). After the docking calculations, we conducted additional 150 ns production simulations on the combined post-docking models, employing the same parameters as described above for the pre-docking production simulations.

## Molecular docking calculations

For each of the four pre-docking, pure-component protein systems, we clustered our initial 1 $\mu$s production simulation trajectory using the gromos method implemented in GROMACS. Representative structures from the top two clusters in each case were selected and prepared for subsequent docking calculations using AutoDock Tools (*Morris et al., 2009*). Docking calculations were carried out with AutoDock Vina (*Trott and Olson, 2010*; *Eberhardt et al., 2021*), treating the proteins as rigid receptors. Given that AutoDock Vina employs a flexible ligand docking procedure, the final PL conformation from each pure-component simulation served as the ligand in the docking calculations. Grids with dimensions of 30 Å × 30 Å × 30 Å were constructed, redundantly covering the complete surface of each receptor protein. The exhaustiveness parameter for docking was set to eight. Docked complexes were analyzed based on the AutoDock score, the number of highly similar complexes, and biological relevance. Unique structures across grids and clusters for each ligand-receptor system were identified for post-docking MD simulations. From molecular docking, we obtained seven unique SOPS/Ig$^{WT}$ models, 5 SOPS/sTREM2$^{WT}$ models, 5 SOPC/Ig$^{WT}$ models, 6 SOPC/sTREM2$^{WT}$ models, 4 SOPS/Ig$^{R47H}$ models, 8 SOPS/sTREM2$^{R47H}$ models, 6 SOPC/Ig$^{R47H}$ models, and 7 SOPC/sTREM2$^{R47H}$ models.

## Trajectory analysis protocols

The Molecular Mechanics Poisson-Boltzmann Surface Area (MM-PBSA) approach was used to calculate PL-protein binding free energies, utilizing the gmx_MMPBSA implementation (*Baker et al., 2001*; *Valdés-Tresanco et al., 2021*; *Miller et al., 2012*). The analysis focused on temporal regions of the simulation trajectories where PL-protein complexes demonstrated stability, as indicated by relatively constant root-mean-square deviation (RMSD) values for at least 50 ns. Over the same temporal regions, we calculated Interaction Entropy, as implemented in gmx_MMPBSA, to estimate entropic trends in PL binding (*Valdés-Tresanco et al., 2021*; *Duan et al., 2016*). Significance testing for these values was performed using two-tailed heteroscedastic T-tests. Fractional occupancy values, characterizing the frequency and location of PL binding on each protein surface, were determined by calculating the fraction of simulation trajectory frames in which a given protein residue was within 4 Å of the PL across all simulations. We calculated conformational changes of the CDR2 loop using MDAnalysis by measuring the distance between residues 45 and 70 at the tops of the CDR1 and CDR2 loops, respectively.

## Results

### The partial stalk domain of sTREM2 differentially modulates the CDR2 loop and broader Ig-like domain dynamics via 'Surface 1' binding in WT and R47H models

Prior TREM2 research identified an open CDR2 loop in R47H models, which disrupts ligand binding to Surface 1 (*Dean et al., 2019*; *Menzies et al., 2019*; *Sudom et al., 2018*; *Lietzke et al., 2025*). To validate our in silico approach, we first sought to recapitulate these findings by performing a 1-μs simulation each of Ig$^{R47H}$ and sTREM2$^{R47H}$. Consistent with prior studies, these 'initial' simulations showed persistently open CDR2 loops in both constructs (*Figure 2—figure supplement 1A*, slate blue and teal lines, respectively; see Methods for loop characterization details). Surprisingly, however—and in contrast to prior experimental studies (*Menzies et al., 2019*; *Sudom et al., 2018*)—a 1-μs simulation each of Ig$^{WT}$ and sTREM2$^{WT}$ (*Figure 2—figure supplement 1A*, pink and red lines, respectively) revealed that the former transitioned from a closed to an open CDR2 loop midway through the simulation.

To understand this unexpected result, we examined whether variation in the protonation state of histidine residue H43 adjacent to the CDR2 loop of Ig$^{WT}$ and Ig$^{R47H}$ could account for the observed differences in CDR2 loop dynamics. Initial protonation states (HSE in Ig$^{WT}$, HDE in Ig$^{R47H}$) were assigned by the pdb2gmx algorithm in GROMACS, based on optimal hydrogen-bonding conformations. Two additional 1-μs simulations of Ig$^{WT}$ were performed with the original protonation state (HSE) and three with the alternate state (HDE), totaling six replicates of Ig$^{WT}$. The open CDR2 loop conformation was sampled in two of the six simulations (*Figure 2—figure supplement 2A-i*, purple and brown lines): the original Ig$^{WT}$ simulation with HSE, and one replicate with HDE. We note that the CDR2 loop remained

open in a 1-μs simulation of Ig$^{R47H}$ with the alternate protonation state (HSE), as well as in four additional replicates performed with the original protonation state (HDE), resulting in six replicates of Ig$^{R47H}$ that sampled an open CDR2 loop (*Figure 2—figure supplement 2B-i*). These results indicate that the protonation state of H43 does not explain the differences in loop dynamics. Rather, the R47H mutation appears to stabilize the open CDR2 loop conformation, whereas loop opening in the WT construct may be more stochastic or transient. These conclusions are further supported by *Figure 2A*, which shows the average CDR2 loop distance and standard error of the mean (SEM) across all six replicates of both Ig$^{WT}$ and Ig$^{R47H}$. This analysis reveals that the average loop distance increased in the second half of the Ig$^{WT}$ simulations, albeit to a lesser extent than in the R47H models.

Notably, sTREM2$^{WT}$ maintained a closed-loop conformation throughout the initial 1-μs simulation (*Figure 2—figure supplement 1A*). As with the Ig constructs, we again performed five additional 1-μs simulations of sTREM2$^{WT}$ (and of sTREM2$^{R47H}$) to further probe this behavior and more robustly sample protein dynamics. Similar to Ig$^{WT}$, an open CDR2 loop conformation was sampled in two of the six sTREM2$^{WT}$ simulations (*Figure 2—figure supplement 2C-i*, yellow and brown lines). However, averaging the loop distance across replicates suggests that CDR2 loop opening occurs with a delay of approximately 200 ns in sTREM2$^{WT}$ compared to Ig$^{WT}$ (*Figure 2A*). This observation hints at a stronger thermodynamic preference for the closed-loop conformation in sTREM2$^{WT}$, potentially due to the presence of the partial stalk domain. In sTREM2$^{WT}$, the stalk may help stabilize the closed state through intramolecular interactions that are sterically hindered in membrane-bound TREM2 (represented by Ig$^{WT}$), where it is tethered to the transmembrane domain and less conformationally flexible—a hypothesis supported by additional structural analyses, described below.

Given the potential stochastic nature of CDR2 loop opening in the absence of the R47H mutation, we sought to examine its impact on protein stability more rigorously using root-mean-square fluctuation (RMSF) analysis. These calculations included the six Ig$^{WT}$ simulations (*Figure 2—figure supplement 2A-ii*) and the isolated Ig-like domain from the six sTREM2$^{WT}$ simulations (sTREM2$^{WT-Ig}$; *Figure 2—figure supplement 2C-ii*), which were subsequently averaged across trajectories for each system (*Figure 2B*). RMSF was calculated relative to the post-minimization structure, which adopted a closed CDR2 loop in WT structures and an open CDR2 loop in R47H structures. On average, sTREM2$^{WT-Ig}$ exhibited slightly reduced, but statistically overlapping, residue-level fluctuations across most of the Ig-like domain compared to Ig$^{WT}$. This suggests that subtle differences in sTREM2$^{WT}$ dynamics may stem from stalk-mediated effects on the sampling of specific conformational states. In light of these findings, we conducted a similar RMSF analysis to compare the dynamics of the six Ig$^{R47H}$ simulations (*Figure 2—figure supplement 2B-ii*) with those of the isolated Ig-like domain from the six sTREM2$^{R47H}$ simulations (sTREM2$^{R47H-Ig}$; *Figure 2—figure supplement 2D-ii*). In contrast to the WT model, the presence of the stalk in sTREM2$^{R47H-Ig}$ led to substantially increased residue-level fluctuations across much of the Ig-like domain, particularly in residues 70–80, which comprise most of the CDR2 loop (*Figure 2B*). Collectively, these findings suggest a dual role for the partial stalk in modulating Ig-like domain dynamics: reducing the likelihood of stochastic CDR2 loop opening in WT constructs, while destabilizing the open loop conformation in the R47H mutant. This stalk-mediated destabilization may contribute to sTREM2's neuroprotective function by preventing prolonged stabilization of the open CDR2 loop state, which is promoted by R47H and may underlie its pathogenic effects.

We next aimed to elucidate the molecular basis of interactions between the Ig-like domain and partial stalk domain in sTREM2. To this end, we calculated the RMSD of Cα atoms across the six replicate simulations for each construct (*Figure 2—figure supplement 2A-D-iii*), tracking temporal deviations relative to the post-equilibration configuration in each case. To highlight specific CDR2 loop dynamics of interest, we compared the RMSD of the initial 1-μs simulations of Ig$^{WT}$ and sTREM2$^{WT}$, and then for Ig$^{R47H}$ and sTREM2$^{R47H}$, as shown in *Figure 2C and D*, respectively. Stalk convergence analyses across all sTREM2 replicates confirmed that these initial simulations captured a dominant high-stability conformation for each construct (*Figure 2—figure supplement 1B*; WT1 (left) and R47H1 (right)). The RMSD of each structure converged relatively quickly during the simulations. Ig$^{WT}$ and Ig$^{R47H}$ displayed behavior consistent with the CDR2 loop distances depicted in *Figure 2A*. Ig$^{R47H}$ exhibited a consistently low RMSD profile throughout the simulation, indicative of a persistently open CDR2 loop (*Figure 2D*, inset **iii**). In contrast, the RMSD profile for Ig$^{WT}$ remained low until approximately 400 ns, at which point it began to increase, signaling a transition from a closed to an open CDR2 loop (*Figure 2C*, insets **iii-iv**). Although the loop remained open for the rest of the simulation, Ig$^{WT}$ showed

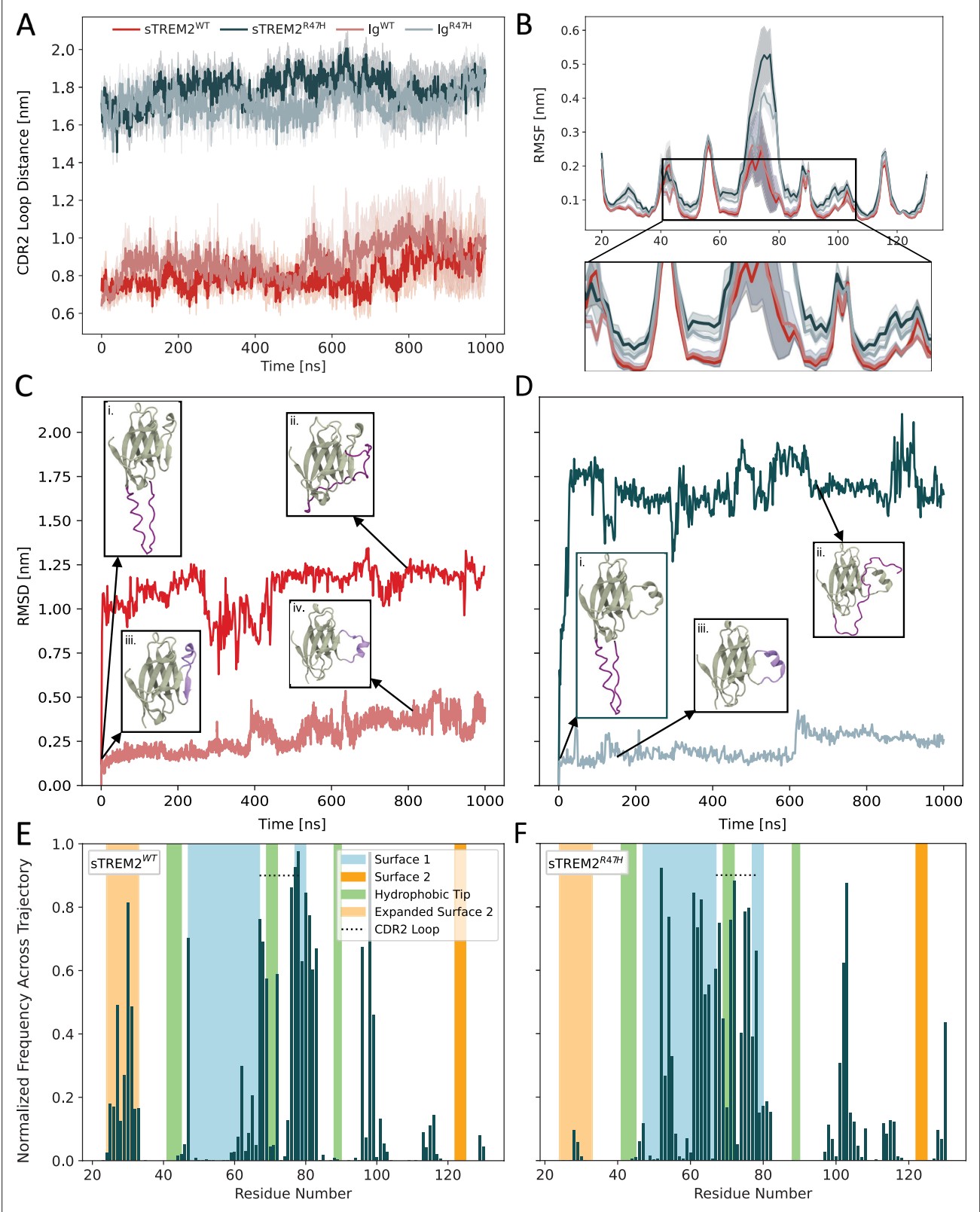

**Figure 2.** The stalk of sTREM2^WT modulates the Ig-like domain via binding to Surface 1. (**A**) Distance between complementarity-determining region (CDR) loop residues 45 and 70 over time for wild-type (WT) and R47H models of soluble form of TREM2 (sTREM2) and TREM2 (Ig), averaged across six replicates. Error bars represent the SEM. (**B**) Temporally averaged Cα root-mean-square fluctuation (RMSF) of the isolated Ig-like domain of WT and R47H sTREM2 compared to the Ig-like domain of TREM2, averaged across six replicates. Error bars represent the SEM. (**C–D**) Cα RMSD of (**C**) WT and

*Figure 2 continued on next page*

*Figure 2 continued*

(**D**) R47H models of sTREM2 and Triggering Receptor Expressed on Myeloid Cells 2 (TREM2) over time during the initial simulations, with corresponding snapshots shown for the starting and equilibrated structures in each case. The stalk domain of sTREM2 is shown in dark purple, and the CDR2 loop of TREM2 is shown in light purple. (**E–F**) Normalized fractional occupancy of residues in the Ig-like domain of (**E**) sTREM2$^{WT}$ and (**F**) sTREM2$^{R47H}$ by the stalk during the initial simulations.

The online version of this article includes the following figure supplement(s) for figure 2:

**Figure supplement 1.** Structural characterization of (s)TREM2 without ligands.

**Figure supplement 2.** Complementarity-determining region (CDR) distance, root-mean-square fluctuation (RMSF), and root-mean-square deviation (RMSD) for (s)TREM2 without ligands.

markedly greater RMSD fluctuations than Ig$^{R47H}$, suggesting that the R47H mutation likely imparts some stability to the open-loop conformation.

Sharp initial increases in RMSD are observed with both sTREM2$^{WT}$ and sTREM2$^{R47H}$. In both cases, the starting structures (inset panels **i** of *Figure 2C–D*) exhibited a generally linear configuration of the stalk, potentially resembling its membrane-bound form. After the initial RMSD increases, the partial stalk domains were observed to consistently interact with the Ig-like domains of sTREM2$^{WT}$ and sTREM2$^{R47H}$ (inset panels **ii** of *Figure 2C–D*). To identify the specific regions of the protein surface with which the partial stalk interacted during the representative simulations, we generated per-residue fractional occupancy maps characterizing the contact frequency (atom-atom distance within 4 Å) between the stalk and each residue in sTREM2$^{WT}$ (*Figure 2E*) and sTREM2$^{R47H}$ (*Figure 2F*). We observed a notable increase in the interaction frequency between residues in Surface 1 and the partial stalk domain of sTREM2$^{R47H}$ compared to sTREM2$^{WT}$. This difference is likely due to the more persistently open CDR2 loop of sTREM2$^{R47H}$, providing greater accessibility of Surface 1 residues for interaction with the stalk. We generated similar occupancy maps, averaged across replicate simulations and observed similar, albeit more diffuse, patterns (*Figure 2—figure supplement 1C–D*).

Given the highly negative overall charge of the stalk domain (–8), its recurrent interactions with the positively charged residues in Surface 1 are perhaps unsurprising. In addition to these contacts, however, the stalk domain in sTREM2$^{WT}$ also frequently interacted with residues 20–35, though typically with lower occupancy. While this region has not been directly noted in previous experimental studies, it lies directly adjacent to residues comprising Surface 2 as defined by *Belsare et al., 2022*. We, therefore, refer to it here as 'Expanded Surface 2.' Taken together, the diversity of surface regions contacted by the stalk domain—and the overlap in many cases with known TREM2 ligand-binding sites—suggests a potential mechanism for differences in ligand binding between sTREM2 and full-length TREM2, which we explore in detail in the following section.

## sTREM2's partial stalk domain modulates PL binding by promoting interactions with a new site on the Ig domain and creating an additional binding site within the stalk itself

In vitro studies examining molecular interactions between (s)TREM2 and ligands remain limited. One study reported that sTREM2 exhibited slightly lower, but statistically insignificant, affinity for monomeric Aβ$_{1-42}$ compared to TREM2 (*Kober et al., 2021*). Based on the similar binding affinities, the authors proposed that the Ig-like domain constitutes the principal binding surface of sTREM2. However, another recent report from the same group found that the TREM2 Ig-like domain does not engage ligands at 'Surface 2' (*Greven et al., 2025*), contrasting earlier cross-linked mass spectrometry data showing Aβ fibrils binding to this region in sTREM2 (*Moutinho and Landreth, 2021*). These conflicting results suggest that differences in ligand binding between sTREM2 and TREM2 likely stem from more complex factors than simply affinity and ligand type. Another study presented a crystal structure of a TREM2 trimer bound by three PS molecules, revealing extensive PS interactions with residues in the CDR2 loop and hydrophobic tip (*Sudom et al., 2018*). While the study did not examine PL/sTREM2 interactions, we hypothesized that similar differential patterns of (s)TREM2 binding may occur with PLs—as previously seen with Aβ—due to stalk-induced alterations in ligand accessibility, which may in turn modulate binding affinity.

To test our hypothesis, we conducted 150-ns simulations of both Ig$^{WT}$ and sTREM2$^{WT}$ bound by the PLs stearoyl-oleoyl-PC (SOPC) and stearoyl-oleoyl-PS (SOPS) in various favorable starting docking

configurations (see Methods). As discussed previously, the initial 1-μs simulation of each protein construct was deemed representative of the replicate-averaged behavior and was thus used for these docking studies. We then analyzed PL binding by calculating the fractional occupancy of each protein residue, temporally averaged across all simulations for each protein/PL system. Across all four systems, the highest occupancy by both PLs was observed at residues within the CDR2 loop and hydrophobic tip (*Figure 3A–D*, primarily green regions), consistent with earlier in vitro findings and supporting the validity of our approach. Nevertheless, distinctions between the SOPS and SOPC models were also evident, underscoring the influence of ligand charge on binding interactions. Specifically, SOPS, being negatively charged, showed higher relative occupancy of the positively charged residues on Surface 1 compared to SOPC, which is neutrally charged (*Figure 3A–B* vs. *Figure 3C-D*, respectively, blue regions). Conversely, SOPC showed higher relative occupancy of residues within the hydrophobic tip than SOPS (*Figure 3C–D* vs. *Figure 3A-B*, green regions).

Upon comparing the fractional occupancy plots of the PL/sTREM2$^{WT}$ simulations (*Figure 3B and D*) with those of the PL/Ig$^{WT}$ simulations (*Figure 3A and C*), we observed a noticeable decrease in the relative occupancy of residues within Surface 1, the hydrophobic tip, and particularly the CDR2 loop in the sTREM2$^{WT}$ systems. Instead, both PL/sTREM2$^{WT}$ simulations showed ligand binding at the newly defined Expanded Surface 2 (*Figure 3B and D*, light orange regions), which was previously shown to interact with the sTREM2$^{WT}$ stalk, albeit with lower occupancy than Surface 1 residues (*Figure 2E*). In contrast, PL binding to Expanded Surface 2 was almost entirely absent in the Ig$^{WT}$ simulations. Collectively, these results suggest that Expanded Surface 2 functions as a secondary ligand-binding surface in sTREM2, becoming preferentially engaged when Surface 1 is occupied by the flexible stalk (and when Expanded Surface 2 is not *itself* engaged by the stalk). Finally, we observed frequent interactions between both PLs and residues within the stalk domain of sTREM2$^{WT}$ (*Figure 3B and D*, pink regions), suggesting that the stalk itself may independently serve as an additional binding site for diverse ligands.

We gained deeper insights into PL/(s) TREM2 interactions through visual analysis of our simulation trajectories using VMD and binding free energy calculations performed with the MM-PBSA approach (see Methods). Representative snapshots from converged portions of the simulations for each protein/PL system (defined by stable RMSD values for at least 50 ns) are shown in *Figure 3E–H*, alongside their corresponding binding free energies. Among the WT systems, a SOPS/Ig$^{WT}$ complex exhibited the most favorable (lowest) binding free energy, consistent with TREM2's known affinity for anionic ligands. Notably, in the lowest-energy binding models, both SOPS and SOPC directly engaged the CDR2 loop in Ig$^{WT}$ (*Figure 3E*, gray model; *Figure 3G*, orange model, respectively). In contrast, both SOPS and SOPC bound most favorably to the newly defined Expanded Surface 2 on sTREM2$^{WT}$ (*Figure 3F*, orange model; *Figure 3H*, purple model). RMSF calculations revealed heightened dynamics in the stalk domain of sTREM2$^{WT}$ for these particular models (*Figure 3—figure supplement 1B-iii and D-iii*), suggesting that PL binding perturbs homeostatic stalk/Ig-like domain interactions. This is further supported by the observation that, in the absence of a ligand, the stalk most frequently interacts with Surface 1 (*Figure 2E*)—the same region where PLs, especially SOPS, bind in the absence of the stalk (*Figure 3A and C*), with reduced occupancy when the stalk is present (*Figure 3B and D*). Together, these results indicate that the PLs compete with the stalk domain for access to the CDR2 loop and Surface 1, likely due to their shared amphipathic and negatively charged character.

To gain deeper insights into the thermodynamics of PL/(s) TREM2 interactions, we performed Interaction Entropy (IE) calculations (see Methods). While many free energy estimates neglect entropy, recent studies have highlighted its essential role in characterizing the full Gibbs free energy landscape in biomolecular systems (*Winkler, 2020*; *Polyansky et al., 2012*). IE calculations capture entropic changes in ligand, protein, and solvent interactions, and efficiently estimate these contributions directly from MD simulations, making them particularly useful for comparing relative entropies (*Duan et al., 2016*). Our results broadly indicate that the entropic contributions (-TΔS) were positive across all systems, with higher values for SOPS models than SOPC models, although the differences were not statistically significant (*Figure 3—figure supplement 2I*). Entropic loss upon ligand binding reflects reduced conformational freedom in the ligand, protein, and/or surrounding solvent (*Duan et al., 2016*; *Winkler, 2020*). Thus, the larger entropic penalties observed for SOPS models suggest greater 'snugness' of binding (*Chang et al., 2007*), implying that binding in these systems is more

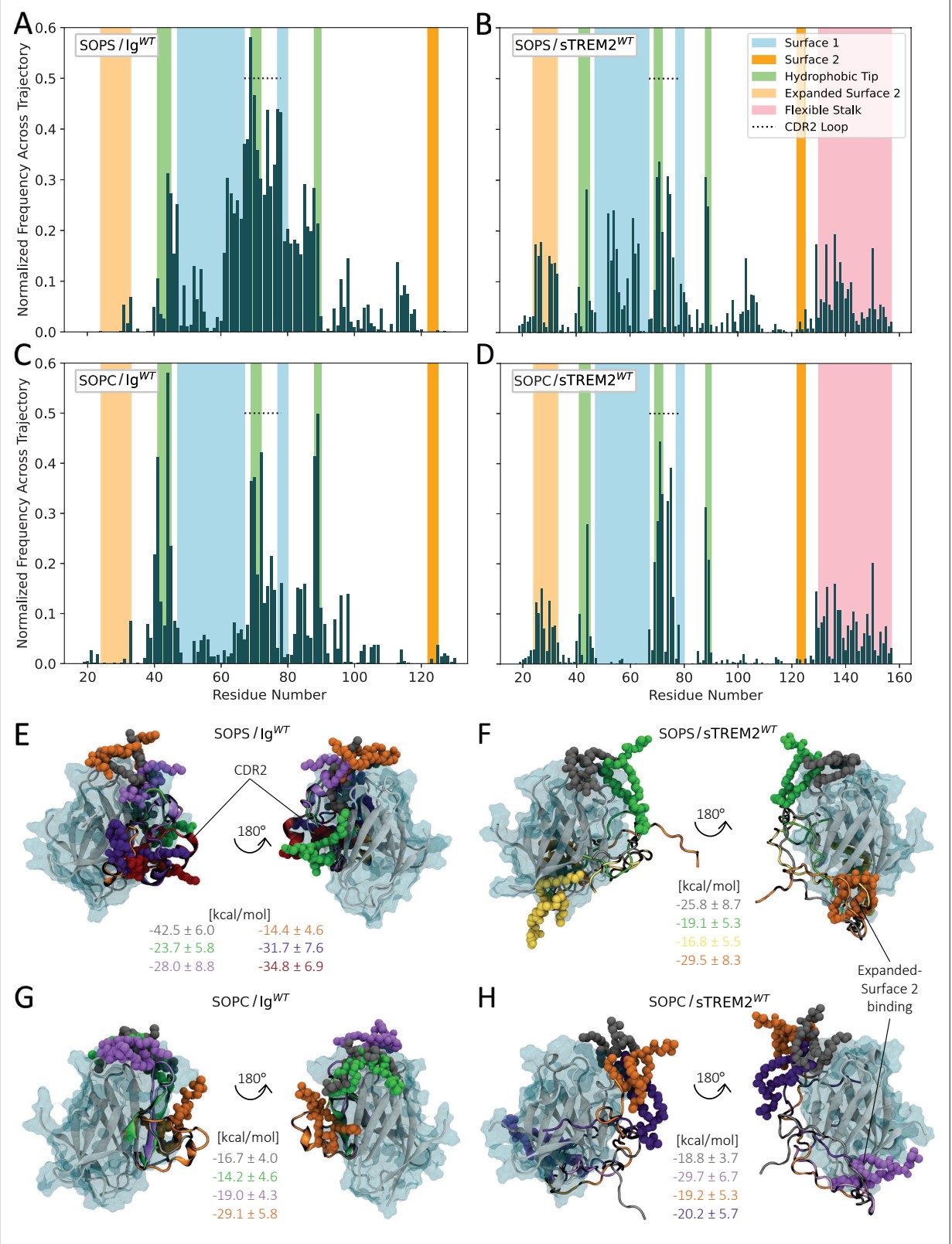

**Figure 3.** The stalk of sTREM2$^{WT}$ modulates phospholipid binding. (**A–D**) Normalized fractional occupancy of residues in (**A**) Ig$^{WT}$ by stearoyl-oleoyl-PS (SOPS), (**B**) sTREM2$^{WT}$ by SOPS, (**C**) Ig$^{WT}$ by stearoyl-oleoyl-PC (SOPC), and (**D**) sTREM2$^{WT}$ by SOPC, averaged across seven, five, five, and six 150-ns simulations, respectively. (**E–H**) Visual Molecular Dynamics (VMD) snapshots of representative structures from the PL/(s)TREM2$^{WT}$ simulations, with phospholipids (PLs) and corresponding complex binding free energies shown in matching colors.

*Figure 3 continued on next page*

*Figure 3 continued*

The online version of this article includes the following figure supplement(s) for figure 3:

**Figure supplement 1.** Root-mean-square deviation (RMSD) and root-mean-square fluctuation (RMSF) analysis for (s)TREM2$^{WT}$ with phospholipids (PLs).

**Figure supplement 2.** Protein-phospholipid (PL) complex renderings and interaction entropy calculations.

enthalpically driven. However, interpretation of these results is limited by large standard deviations, likely due to the highly dynamic nature of PL/(s) TREM2 interactions (*Ekberg and Ryde, 2021*).

Finally, some PL/TREM2 complexes were observed to undergo rapid conformational changes in the CDR2 loop during the simulations, indicated by marked increases in the RMSD of the protein (*Figure 3—figure supplement 1A-i,C-i*) combined with high RMSF values in CDR2 loop residues (*Figure 3—figure supplement 1A-iii,C-iii*). These changes—likely facilitated by elevated, direct PL binding to the CDR2 loop of Ig$^{WT}$ compared to sTREM2$^{WT}$, as discussed earlier—occurred on much shorter timescales than in the ligand-free simulations and included dynamic opening and closing of the loop along with shifts in its $\alpha$-helical character. This suggests a mechanism by which the CDR2 loop may function as a dynamically responsive element during ligand binding. However, further experimental studies are needed to clarify the functional significance of CDR2 loop remodeling in response to ligand binding and to define the conditions under which such conformational transitions occur.

## The AD-risk mutation R47H diminishes ligand discrimination by and binding to the Ig-like domain of (s)TREM2, while sTREM2's stalk domain partially restores overall ligand binding

To investigate the roles of (s) TREM2 in disease pathology, we examined PL binding in the presence of the AD-risk mutation R47H. Previous studies have suggested that the decreased ligand-binding capabilities of TREM2$^{R47H}$ may underlie its observed loss-of-function (*Kober et al., 2016*; *Kober et al., 2021*; *Sudom et al., 2018*; *Wang et al., 2015*). Notably, one study showed reduced reporter cell activity of TREM2$^{R47H}$ compared to TREM2$^{WT}$ when binding anionic lipids such as PS, while no significant difference was observed in their binding to PC (*Wang et al., 2015*). Separately, sTREM2$^{R47H}$ and sTREM2$^{WT}$ were shown to bind Aβ and inhibit fibrillization to a similar extent (*Belsare et al., 2022*). Taken together, these studies suggest that the effects of R47H extend beyond reduced binding affinity and may also be influenced by ligand type, binding site, membrane context, and the presence of the stalk.

Upon examining the fractional occupancy plots of the PL/(s) TREM2$^{R47H}$ simulations, we observed broadly similar patterns of binding to the Ig-like domain, regardless of stalk presence or PL charge (*Figure 4A–D*). Across all four simulations, PLs predominantly occupied the CDR2 loop and Surface 1 of the Ig-like domain. This uniformity in binding contrasts sharply with the previously discussed behavior in the PL/(s) TREM2$^{WT}$ simulations (*Figure 3A–D*), where distinct differences were observed between SOPS and SOPC binding to both Ig$^{WT}$ and sTREM2$^{WT}$, as well as between the two proteins for a given PL. These results suggest that the presence of R47H diminishes the ability of Ig$^{R47H}$ and sTREM2$^{R47H}$ to distinguish between crucial brain-derived ligands, proposing a novel mechanism of loss-of-function for this key AD-risk mutation.

Compared to the PL/(s)TREM2$^{WT}$ simulations, all four PL/(s)TREM2$^{R47H}$ simulations demonstrated markedly reduced PL binding to the hydrophobic tip. These reductions likely stem from conformational effects induced by R47H, which promotes a more open and rigid CDR2 loop adjacent to the hydrophobic tip, thereby altering its accessibility and ligand-binding properties. In addition, changes in stalk-Ig-like domain interactions observed in the presence of R47H (*Figure 2E–F*) may further influence PL binding by reshaping the local structural landscape. Consistent with these effects, we also observed diminished PL binding to Expanded Surface 2 in the sTREM2$^{R47H}$ simulations, suggesting that R47H not only impairs canonical ligand recognition surfaces but may also disrupt secondary binding modes unique to sTREM2.

Compensating for these reductions in binding, increased PL binding to Surface 1 was observed in all four R47H simulations, along with enhanced PL binding to the flexible stalk of sTREM2$^{R47H}$. These results provide further support for the notion that sTREM2's stalk domain functions as an independent ligand-binding site. Moreover, they suggest the stalk domain may possess an inherent capacity to 'rescue' sTREM2 deficiencies in ligand binding to the Ig-like domain caused by disease-associated

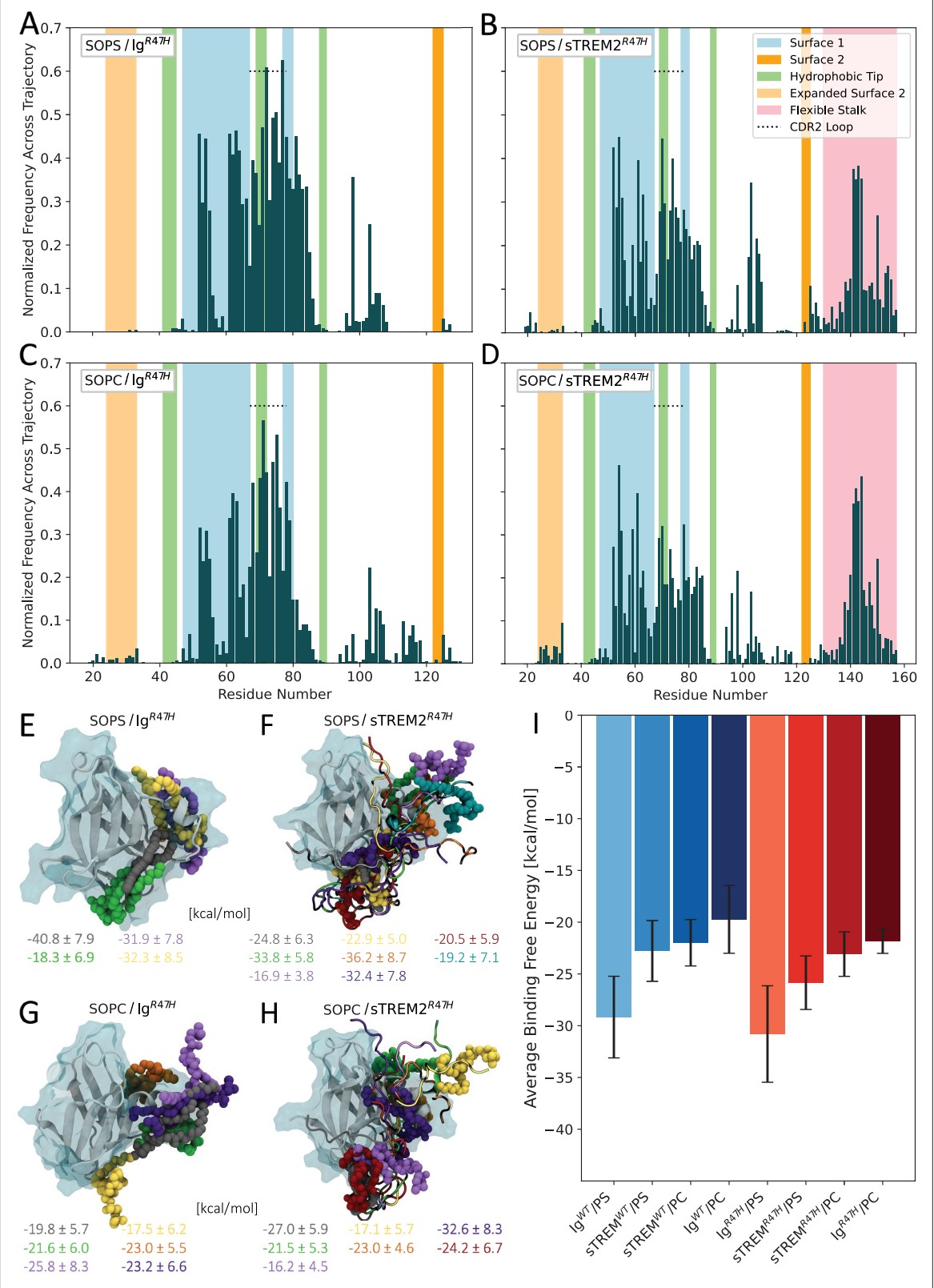

**Figure 4.** The R47H mutation decreases ligand discrimination but increases binding to the flexible stalk domain. (**A–D**) Normalized fractional occupancy of residues in (**A**) Ig$^{R47H}$ by stearoyl-oleoyl-PS (SOPS), (**B**) sTREM2$^{R47H}$ by SOPS, (**C**) Ig$^{R47H}$ by stearoyl-oleoyl-PC (SOPC), and (**D**) sTREM2$^{R47H}$ by SOPC, averaged across four, seven, six, and eight 150-ns simulations, respectively. (**E–H**) Visual Molecular Dynamics (VMD) snapshots of representative structures from the PL/(s) TREM2$^{R47H}$ simulations, with phospholipids (PLs) and corresponding complex binding free energies shown in matching colors.

*Figure 4 continued on next page*

*Figure 4 continued*

(I) Comparison of the binding free energies averaged across all WT and R47H models for each PL/protein system. Error bars represent the standard error of the mean.

The online version of this article includes the following figure supplement(s) for figure 4:

**Figure supplement 1.** Root-mean-square deviation (RMSD) and root-mean-square fluctuation (RMSF) analysis for (s) TREM2$^{R47H}$ with phospholipids (PLs).

mutations, offering a mechanistic explanation for sTREM2's neuroprotective role in AD. Indeed, this may help explain why, as noted earlier, R47H does not significantly impair sTREM2$^{R47H}$'s ability to bind Aβ and inhibit fibrillization to the same extent as sTREM2$^{WT}$.

Following the methodologies used to evaluate the PL/(s)TREM2$^{WT}$ simulations, we conducted visual analysis of our trajectories paired with binding free energy calculations. Representative structures with corresponding binding free energies are shown in ***Figure 4E–H***. Broadly, the trends align closely with those observed for the WT models (***Figure 3E–H***), with a SOPS/Ig$^{R47H}$ complex again exhibiting the lowest (most favorable) binding free energy (***Figure 4E***). Unlike the WT models, however, these results do not indicate competitive binding between the PLs and the stalk, as evidenced by high PL interaction frequencies with the CDR2 loop and Surface 1 that persist even when the stalk is present (***Figure 4B and D*** vs. ***Figure 4A,C***), despite the stalk's increased interaction frequency with Surface 1 in the context of R47H (***Figure 2F*** vs. ***Figure 2E***). This is likely due to the more persistently open CDR2 loop in the R47H complexes, which alters surface accessibility and stalk dynamics.

Our binding free energy results show that across all models, sTREM2 and TREM2 bind PS more favorably than PC (***Figure 4I***), affirming previous findings that TREM2 favors anionic ligands (***Kober and Brett, 2017***). However, these differences were not statistically significant. We also observed no significant difference or trend in binding free energies between PS/(s)TREM2$^{WT}$ and PS/(s)TREM2$^{R47H}$ complexes, contrasting with previous experimental findings that R47H reduces TREM2's affinity for endogenous ligands (***Kober et al., 2021***; ***Sudom et al., 2018***; ***Wang et al., 2015***). Similar to WT models, SOPS binding poses in the R47H models exhibited higher average entropic contributions than their SOPC counterparts, with statistically significant differences for sTREM2$^{R47H}$ but not for Ig$^{R47H}$ (***Figure 3—figure supplement 2I***). Together, these observations suggest that reduced ligand affinity alone does not fully account for the signaling deficits associated with the R47H variant. More broadly, our results indicate that differences in binding across a range of ligands, (s)TREM2 variants, and between sTREM2 and TREM2 are likely driven by differences in accessibility of protein surface residues and ligand binding patterns, factors that extend beyond mere differences in affinity.

Lastly, we observed a prominent difference in the dynamic behavior of the CDR2 loop between Ig$^{R47H}$ and Ig$^{WT}$. In most simulations, the CDR2 loop in Ig$^{R47H}$ remained stably open regardless of PL binding (***Figure 4—figure supplement 1***), whereas in Ig$^{WT}$, the loop underwent dynamic opening and closing in response to ligand engagement. These findings suggest that, beyond impairing ligand affinity and selectivity, the R47H mutation may also hinder the CDR2 loop's ability to respond adaptively to ligand binding, potentially disrupting physiological TREM2 signaling. Future investigation of the TREM2-DAP12 complex will be essential to clarify how ligand binding to the Ig-like domain propagates downstream signaling, and how this process may be disrupted by the R47H variant.

## Discussion
### Main conclusions
We utilized long-timescale MD simulations to investigate relevant structural domains of sTREM2 and TREM2, along with their interactions with key PLs in the brain. Through the analysis of RMSF, RMSD, and PL/protein residue occupancy calculations, we established the flexible stalk domain of sTREM2 as the molecular basis for differential ligand binding between sTREM2 and TREM2. This difference in binding arises from the stalk's role in modulating dynamics of the Ig-like domain and altering the ligand accessibility of its surface residues. By integrating free energy, interaction entropy, and ligand occupancy calculations, we quantified the energetics of these interactions, confirming the presence of an alternate ligand binding site on sTREM2$^{WT}$, which we termed the 'Expanded Surface 2.' Binding of PLs to this site arises from competitive binding of the flexible stalk to Surface 1 on the Ig-like domain.

These stalk-Ig-like domain interactions were disrupted in the presence of the AD-risk mutation R47H and entirely absent in the Ig (TREM2) models, indicating occupancy of Expanded Surface 2 occurs solely with sTREM2$^{WT}$. These observations underscore our conclusion that, rather than (or in addition to) sTREM2's previously conceived role as a dummy receptor for TREM2, the flexible stalk confers sTREM2 with unique ligand binding preferences and patterns that facilitate distinct endogenous functions compared to TREM2.

Furthermore, we found that the stalk domain itself serves as an independent site for ligand binding, with heightened PL occupancy observed in the presence of R47H compared to the wild-type model. This suggests that the stalk domain may have the capacity to partially 'rescue' dysfunctional ligand binding to the Ig-like domain of sTREM2 caused by disease-associated mutations like R47H. Moreover, our observations for both sTREM2 and TREM2 indicate that R47H-induced dysfunction may result not only from diminished ligand binding but also an impaired ability to discriminate between different ligands in the brain, proposing a novel mechanism for loss-of-function. In summary, the findings of this study reveal the endogenous structural and dynamical mechanisms of (s)TREM2, a critical component in AD pathology. These insights offer new fundamental knowledge that can serve as guiding principles for the design of future therapeutics, paving the way for potential advancements in AD treatment strategies.

## Ideas and speculation

Our findings suggest that loss-of-function in sTREM2 and TREM2 occurs not only through reductions in ligand binding affinity, but also through a change in ligand binding patterns and a loss of ligand discrimination capacity. Altered ligand binding may hinder the ability of TREM2 and its co-signaling partner DAP12 to transmit signals across the cell membrane. This deficiency implicates an impaired microglial response to ligand binding as a key mechanism for dysfunction in AD. Pathologically, this would lead to reduced lipid uptake (*McQuade et al., 2020*), diminished Aβ plaque clearance (*Zhao et al., 2018*), decreased microglial activation (*McQuade et al., 2020*), and inhibited intracellular lipid metabolism (*Wei et al., 2024*). Previous studies speculate that reduced microglial lipid metabolism can trigger neurotoxic activation states or loss of neuroprotective functions (*Damisah et al., 2020*). Furthermore, the inability to discriminate among PLs may result in an invariable response from microglia when presented with diverse ligands. While PC is the most abundant PL in cell membranes, PS expression on the outer membrane leaflet increases in apoptotic cells, acting as a damage-associated signal (*Nagata et al., 2016*). Failure to differentiate between these PLs may lead to chronic over-activation of microglia, or at the very least, loss of endogenous protective functions. Additionally, there are two known alternatively spliced isoforms of sTREM2 which constitute the minority of CSF sTREM2 levels. These isoforms occur due to alternate splice sites and skipping within exon 4 (*Moutinho and Landreth, 2021*). While our study suggests that the flexible stalk domain of sTREM2 is highly relevant, the specific conclusions do not extend to these isoforms, as they have entirely different stalk compositions and lengths. A separate study of these isoforms is pertinent.

Recently, monoclonal antibodies targeting TREM2 have emerged (*Fassler et al., 2023*; *Schlepckow et al., 2020*; *van Lengerich et al., 2023*), designed to bind to the stalk region above its cleavage site (*Schlepckow et al., 2020*; *van Lengerich et al., 2023*). Consequently, the primary binding epitope of at least one of these antibodies also resides on sTREM2. Notably, this antibody also reduces the shedding of sTREM2 (*van Lengerich et al., 2023*). It is conceivable that treatment with this and similar antibodies may compromise the endogenous function of sTREM2, given that the stalk, a domain our study has identified as functionally significant, may be sequestered by the antibody. This may ultimately result in off-target effects for these therapeutics and prompts a consideration of how to separately target sTREM2 and TREM2.

## Limitations

As with all models, particularly computational ones, it is crucial to recognize their limitations. In our study, the initial positions of PLs bound to (s)TREM2 were determined exclusively by AutoDock Vina. To mitigate potential bias from the docking process, we conducted MD simulations on all unique structures observed in the top 20 docked models. Rigorous, yet computationally efficient free energy calculations remain a challenge for computational protein-ligand studies. Despite employing the latest MM-PBSA free energy calculation algorithms and novel interaction entropy calculations, achieving

statistical significance for many of the trends we observed that align with experiments remained challenging. This difficulty was further compounded by our approach of averaging statistics across multiple models of PLs docked to different initial binding locations on (s)TREM2. While this approach was intended to holistically capture PL/(s)TREM2 interactions, it effectively diluted our sample size against any specific epitope, potentially impacting the statistical significance of our findings. Future studies could address these challenges by utilizing larger sample sizes of independent MD simulations and more intensive energy calculations as computational power and resources continue to advance.

Moreover, our study, like previous investigations of TREM2, partially assumed that the Ig-like domain is representative of membrane-bound TREM2. This assumption overlooks potential contributions from the membrane, membrane-bound flexible stalk domain, DAP12, or TREM2 glycosylation. Ultimately, more biologically representative simulations of membrane-bound TREM2 are warranted to test this assumption. Additionally, it is unlikely that PLs are presented as individual soluble ligands in the brain in an in vivo setting. Instead, they are more likely co-presented with other components in lipoproteins or in contexts such as demyelination, cell debris, or Aβ. This co-presentation would influence the charge and steric profiles of lipid presentation. While our study provides valuable insights and fills existing knowledge gaps, further research is necessary to fully understand TREM2 binding in more biologically relevant ligand-binding contexts.

## Acknowledgements

This work utilized the Alpine high-performance computing resource and the Blanca condo computing resource at the University of Colorado Boulder. Alpine is jointly funded by the University of Colorado Boulder, the University of Colorado Anschutz, and Colorado State University. Blanca is jointly funded by computing users and the University of Colorado Boulder. This material is based upon work supported by the National Science Foundation Graduate Research Fellowship Program under Fellow ID (2024361899). Any opinions, findings, conclusions, or recommendations expressed in this material are those of the author(s) and do not necessarily reflect the views of the National Science Foundation.

## Additional information

### Funding

| Funder | Grant reference number | Author |
| --- | --- | --- |
| National Science Foundation | 2024361899 | David Saeb |

The funders had no role in study design, data collection and interpretation, or the decision to submit the work for publication.

### Author contributions

David Saeb, Conceptualization, Data curation, Software, Formal analysis, Funding acquisition, Validation, Investigation, Visualization, Methodology, Writing – original draft, Project administration, Writing – review and editing; Emma E Lietzke, Conceptualization, Supervision, Project administration, Writing – review and editing; Daisy I Fuchs, Supervision, Project administration; Emma C Aldrich, Visualization, Writing – review and editing; Kimberley D Bruce, Funding acquisition, Writing – original draft, Writing – review and editing; Kayla G Sprenger, Conceptualization, Resources, Data curation, Software, Formal analysis, Supervision, Funding acquisition, Validation, Investigation, Visualization, Methodology, Writing – original draft, Project administration, Writing – review and editing

### Author ORCIDs

David Saeb http://orcid.org/0000-0003-3332-7491
Emma E Lietzke http://orcid.org/0009-0005-5040-418X
Daisy I Fuchs http://orcid.org/0009-0006-5568-5045
Emma C Aldrich http://orcid.org/0000-0002-5729-4738
Kayla G Sprenger https://orcid.org/0000-0002-7505-7920

Reviewer #1 (Public review): https://doi.org/10.7554/eLife.102269.3.sa1
Reviewer #2 (Public review): https://doi.org/10.7554/eLife.102269.3.sa2
Author response https://doi.org/10.7554/eLife.102269.3.sa3

## Additional files

### Supplementary files
MDAR checklist

### Data availability
The current manuscript is a computational study, so no experimental data have been generated for this manuscript. All analyzed computational data is included in the main figures and supplementary figures. Processed trajectories for production simulations and related structure files are available on Dryad at the following link: https://doi.org/10.5061/dryad.0000000gn.

The following dataset was generated:

| Author(s) | Year | Dataset title | Dataset URL | Database and Identifier |
|---|---|---|---|---|
| Saeb D, Lietzke E, Fuchs D, Aldrich E, Bruce K, Sprenger K | 2025 | The flexible stalk domain of sTREM2 modulates its interactions with brain-based phospholipids | https://doi.org/10.5061/dryad.0000000gn | Dryad Digital Repository, 10.5061/dryad.0000000gn |

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
