## [Editor Report · eLife Assessment]

This **useful** manuscript addresses some key molecular mechanisms on the neuroprotective roles of soluble TREM2 in neurodegenerative diseases. The study will advance our understanding of TREM2 mutations, particularly on the damaging effect of known TREM2 mutations, and also provides **solid** evidence why soluble TREM2 can antagonize Aβ aggregation.

---

## [Referee Report · Reviewer #1 (Public review)]

In this manuscript, Saeb et al reported the mechanistic roles of the flexible stalk domain in sTREM2 function using molecular dynamics simulations. They have reported some interesting molecular bases explaining why sTREM2 shows protective effects during AD, such as partial extracellular stalk domain promoting binding preference and stabilities of sTREM2 with its ligand even in the presence of known AD-risk mutation, R47H. Furthermore, they found that the stalk domain itself acts as the site for ligand binding by providing an "expanded surface", known as 'Expanded Surface 2' together with the Ig-like domain. Also, they observed no difference in the binding free energy of phosphatidyl-serine with wild TREM2-Ig and mutant TREM2-Ig, which is a bit inconsistent with the previous report with experiment studies by Journal of Biological Chemistry 293, (2018), Alzheimer's and Dementia 17, 475-488 (2021), Cell 160, 1061-1071 (2015).

---

## [Referee Report · Reviewer #2 (Public review)]

Significance:

TREM2 is an immunomodulatory receptor expressed on myeloid cells and microglia in the brain. TREM2 consists of a single immunoglobular (Ig) domain that leads into a flexible stalk, transmembrane helix, and short cytoplasmic tail. Extracellular proteases can cleave TREM2 in its stalk and produce a soluble TREM2 (sTREM2). TREM2 is genetically linked to Alzheimer's disease (AD), with the strongest association coming from an R47H variant in the Ig domain. Despite intense interest, the full TREM2 ligand repertoire remains elusive, and it is unclear what function sTREM2 may play in the brain. The central goal of this paper is to assess the ligand-binding role of the flexible stalk that is generated during the shedding of TREM2. To do this, the authors simulate the behavior of constructs with and without stalk. However, it is not clear why the authors chose to use the isolated Ig domain as a surrogate for full-length TREM2. Additionally, experimental binding evidence that is misrepresented by the authors contradicts the proposed role of the stalk.

Summary and strengths:

The authors carry out MD simulations of WT and R47H TREM2 with and without the flexible stalk. Simulations are carried out for apo TREM2 and for TREM2 in complex with various lipids. They compare results using just the Ig domain to results including the flexible stalk that is retained following cleavage to generate sTREM2. The computational methods are well-described and should be reproducible. The long simulations are a strength, as exemplified in Figure 2A where a CDR2 transition happens at ~400-600 ns. The stalk has not been resolved in structural studies, but the simulations suggest the intriguing and readily testable hypothesis that the stalk interacts with the Ig domain and thereby contributes to the stability of the Ig domain and to ligand binding. I suspect biochemists interested in TREM2 will make testing this hypothesis a high priority.

Comments on latest version:

The authors have addressed my critiques and carried out additional simulations, as requested. I would upgrade my assessment of the evidence to "solid."

---

## [Author Response]

The following is the authors’ response to the original reviews

**Public Review:**

**Review #1 (Public review):**
Also, they observed no difference in the binding free energy of phosphatidyl-serine with wild TREM2-Ig and mutant TREM2-Ig, which is a bit inconsistent with the previous report with experiment studies by Journal of Biological Chemistry 293, (2018), Alzheimer's and Dementia 17, 475-488 (2021), Cell 160, 1061-1071 (2015).

We agree with the reviewer that our results do not fully recapitulate experimental findings and directly note this in the body of our work, particularly given the known limitations of free energy calculations in MD simulations, as outlined in the Limitations section. Our claim is that the loss-of-function effects of the R47H variant extend beyond decreased binding affinities which are likely due to variable binding patterns. We have also re-analyzed and highlighted statistically significant differences in interaction entropies. Ultimately, our claim is that mutational effects extend beyond experimentally confirmed differences in binding affinities.

Perhaps the authors made significant efforts to run a number of simulations for multiple models, which is nearly 17 microseconds in total; none of the simulations has been repeated independently at least a couple of times, which makes me uncomfortable to consider this finding technically true. Most of the important conclusions that authors claimed, including the opposite results from previous research, have been made on the single run, which raises the question of whether this observation can be reproduced if the simulation has been repeated independently. Although the authors stated the sampling number and length of MD simulations in the current manuscript as a limitation of this study, it must be carefully considered before concluding rather than based on a single run.

To address this comment, we have added numerous replicates to our simulations of WT and R47H (s) TREM2 without lipids and substantially increased the total simulation time. Each pure protein system now has six total microsecond-long technical replicates. The addition of replicates strengthens the validity of the work and allows us to make stronger novel conclusions than with one simulation alone, particularly for claims regarding the CDR2 loop and sTREM2 stalk. In our models with phospholipids, running multiple independent biological replicates of the same system offers a more rigorous methodology than simply repeating simulations of the same docked model. This strategy allows us to sample several distinct starting configurations, thereby minimizing biases introduced by docking algorithms and single-model reliance.

sTREM2 shows a neuroprotective effect in AD, even with the mutations with R47H, as evidenced by authors based on their simulation. sTREM2 is known to bind Aβ within the AD and reduce Aβ aggregation, whereas R47H mutant increases Aβ aggregation. I wonder why the authors did not consider Aβ as a ligand for their simulation studies. As a reader in this field, I would prefer to know the protective mechanism of sTREM2 in Aβ aggregation influenced by the stalk domain.

Our initial approach for this study used Aβ as a ligand rather than phospholipids. However, we noted the difficulties in simulating Aβ, particularly in choosing relevant Aβ structures and oligomeric states (n-mers). We believe that phospholipids represent an equally pertinent ligand for TREM2, given its critical role in lipid sensing and metabolism. Furthermore, there is growing recognition in the AD research community of the need to move beyond Aβ and focus on other understudied pathological mechanisms.

In a similar manner, why only one mutation is considered "R47H" for the study? There are more server mutations reported to disrupt tethering between these CDRs, such as T66M. Although this "T66M" is not associated with AD, I guess the stalk domain protective mechanism would not be biased among different diseases. Therefore, it would be interesting to see whether the findings are true for this T66M.In most previous studies, the mechanism for CDR destabilization by mutant was explored, like the change of secondary structures and residue-wise interloop interaction pattern. While this is not considered in this manuscript, neither detailed residue-wise interaction that changed by mutant or important for 'ligand binding" or "stalk domain".

These are both excellent points that deserve extensive investigation, although we note that our paper does include significant protein-protein and protein-ligand interaction mapping that encompasses both the CDR2 loop and stalk, analyses which were not performed in any previous papers. In a separate paper, we explored more detailed residue-wise interactions for the CDR2 loop (Lietzke et al., Alzheimer’s and Dementia, 2025). While R47H is the most common and prolific mutation in literature, an extensive catalog of other mutations is important to explore. To this end, we are currently preparing a separate publication that will explore a larger mutational library and include more detailed sTREM2 analyses.

The comparison between the wild and mutant and other different complex structures must be determined by particular statistical calculations to state the observed difference between different structures is significant. Since autocorrelation is one of the major concerns for MD simulation data for predicting statistical differences, authors can consider bootstrap calculations for predicting statistical significance.

The addition of numerous replicates across systems negates potential effects from autocorrelation and allows us to include standard deviations to critically assess the validity of our claims.

**Review #2 (Public review):**
The authors state that reported differences in ligand binding between the TREM2 and sTREM2 remain unexplained, and the authors cite two lines of evidence. The first line of evidence, which is true, is that there are differences between lipid binding assays and lipid signaling assays. However, signaling assays do not directly measure binding. Secondly, the authors cite Kober et al 2021 as evidence that sTREM2 and TREM2 showed different affinities for Abeta1-42 in a direct binding assay. Unfortunately, when Kober et al measured the binding of sTREM2 and Ig-TREM2 to Abeta they reported statistically identical affinities (Kd = 3.8 {plus minus} 2.9 µM vs 5.1 {plus minus} 3.7 µM) and concluded that the stalk did not contribute measurably to Abeta binding.

We appreciate the reviewer’s insight and acknowledge the need to clarify our interpretation of Kober et al. (2021). We have adjusted how we cite Kober et al. and reframed the first paragraph in the second results section.

In line with these findings, our energy calculations reveal that sTREM2 exhibits weaker—but still not statistically significant—binding affinities for phospholipids compared to TREM2. These results suggest that while overall binding affinity might be similar, differences in binding patterns or specific lipid interactions could still contribute to functional differences observed between TREM2 and sTREM2.

The authors appear to take simulations of the Ig domain (without any stalk) as a surrogate for the full-length, membrane-bound TREM2. They compare the Ig domain to a sTREM2 model that includes the stalk. While it is fully plausible that the stalk could interact with and stabilize the Ig domain, the authors need to demonstrate why the full-length TREM2 could not interact with its own stalk and why the isolated Ig domain is a suitable surrogate for this state.

We believe that this is a major limitation of all computational work of TREM2 to-date, and of experimental work which only presents the Ig-like domain. This is extensively discussed in the limitations section of our paper and treated carefully throughout the text. We are currently working toward a separate manuscript that will represent the first biologically relevant model of full-length TREM2 in a membrane and will rigorously assess the current paradigm of using the Ig-like domain as an experimental surrogate for TREM2.

**Recommendations for the authors:**

**Reviewer #1 (Recommendations for the authors):**
(1) Perhaps the authors made significant efforts to run a number of simulations for multiple models, which is nearly 17 microseconds in total; none of the simulations has been repeated independently at least a couple of times, which makes me uncomfortable to consider this finding technically true. Most of the important conclusions that authors claimed, including the opposite results from previous research, have been made on the single run, which raises the question of whether this observation can be reproduced if the simulation has been repeated independently. Although the authors stated the sampling number and length of MD simulations in the current manuscript as a limitation of this study, it must be carefully considered before concluding rather than based on a single run.

To address this comment, we have added numerous replicates to our simulations of WT and R47H (s)TREM2 without lipids and substantially increased the total simulation time. Each pure protein system now has six total microsecond-long technical replicates. The addition of replicates strengthens the validity of the work and allows us to make stronger novel conclusions than with one simulation alone, particularly for claims regarding the CDR2 loop and sTREM2 stalk. In our models with phospholipids, running multiple independent biological replicates of the same system offers a more rigorous methodology than simply repeating simulations of the same docked model. This strategy allows us to sample several distinct starting configurations, thereby minimizing biases introduced by docking algorithms and single-model reliance.

(2) sTREM2 shows a neuroprotective effect in AD, even with the mutations with R47H, as evidenced by authors based on their simulation. sTREM2 is known to bind Aβ within the AD and reduce Aβ aggregation, whereas R47H mutant increases Aβ aggregation. I wonder why the authors did not consider Aβ as a ligand for their simulation studies. As a reader in this field, I would prefer to know the protective mechanism of sTREM2 in Aβ aggregation influenced by the stalk domain.

Our initial approach for this study used Aβ as a ligand rather than phospholipids. However, we noted the difficulties in simulating Aβ, particularly in choosing relevant Aβ structures and oligomeric states (n-mers). We believe that phospholipids represent an equally pertinent ligand for TREM2, given its critical role in lipid sensing and metabolism. Furthermore, there is growing recognition in the AD research community of the need to move beyond Aβ and focus on other understudied pathological mechanisms.

(3) In a similar manner, why only one mutation is considered "R47H" for the study? There are more server mutations reported to disrupt tethering between these CDRs, such as T66M. Although this "T66M" is not associated with AD, I guess the stalk domain protective mechanism would not be biased among different diseases. Therefore, it would be interesting to see whether the findings are true for this T66M.(4) In most previous studies, the mechanism for CDR destabilization by mutant was explored, like the change of secondary structures and residue-wise interloop interaction pattern. While this is not considered in this manuscript, neither detailed residue-wise interaction that changed by mutant or important for 'ligand binding" or "stalk domain".

These are both excellent points that deserve extensive investigation, although we note that our paper does include significant protein-protein and protein-ligand interaction mapping that encompasses both the CDR2 loop and stalk, analyses which were not performed in any previous papers. In a separate paper, we explored more detailed residue-wise interactions for the CDR2 loop (Lietzke et al., Alzheimer’s and Dementia, 2025). While R47H is the most common and prolific mutation in literature, an extensive catalog of other mutations is important to explore. To this end, we are currently preparing a separate publication that will explore a larger mutational library and include more detailed sTREM2 analyses.

(5) The comparison between the wild and mutant and other different complex structures must be determined by particular statistical calculations to state the observed difference between different structures is significant. Since autocorrelation is one of the major concerns for MD simulation data for predicting statistical differences, authors can consider bootstrap calculations for predicting statistical significance.

The addition of numerous replicates across systems negates potential effects from autocorrelation and allows us to include standard deviations to critically assess the validity of our claims.

**Reviewer #2 (Recommendations for the authors):**
Major points:(1) I encourage the authors to review Figure 5D and the text of section 2.7 from Kober et al 2021, which argued that "(t)he identical (within error) binding affinities indicated that the TREM2 Ig domain composes the majority (if not entirety) of the mAβ42 binding surface."

We appreciate the reviewer’s insight and acknowledge the need to clarify our interpretation of Kober et al. (2021). We have adjusted how we cite Kober et al and reframed the first paragraph in the second results section.

(2) The abstract and text need extensive revision to address the major concerns, which jeopardize the biological premise and significance of the work.

We have made changes to the abstract and text to reflect concerns and revisions.

(3) The title and abstract should change to reflect the contents of the paper. The authors do not directly measure lipid binding, nor are any of the computations done in a membrane environment. The authors do not measure anything in the brain.

We have modified the title to better reflect the content of the paper. The paper measures lipid binding in the form of free energy calculations and interaction maps.

Minor points:(1) How does the conservation of the TREM2 stalk compare to the Ig domain as they relate to the TREM2 family?

While this study may inspire further exploration of other TREM receptors, we do not believe that our results extend to other TREM family members because of relatively low homology.

(2) Please show the locations of the glycosylation sites on a model in Figure 1 and discuss their potential contribution to the ligand binding surfaces.

N-linked glycosylation points are now noted on the sequence map of Figure 1 and updated in the text.

(3) There is an isoform of TREM2 that produces a secreted product that is similar to the sTREM2 produced by proteolysis. The authors should comment as to whether their findings would apply to secreted TREM2.

We have addressed this with a new line in the ‘Ideas and Speculation’ section.

(4) This sentence on p. 2, line 73 references a review, not a study:

This has been corrected.

(5) "Yet, one study suggested effective TREM2 stimulation by PLs may require co-presentation with other molecules, potentially reflecting the nature of lipoprotein endocytosis30"

This has been corrected.

(6) Is "inclusive" on line 88 a typo for inconclusive?

This has been corrected.

(7) "Further, there is a strong correlation between the levels of sTREM2 in the cerebrospinal fluid and that of Tau, however correlation with Aβ is inclusive"

This has been corrected.